

# Self-affine subglacial roughness: consequences for radar scattering and basal thaw discrimination in northern Greenland

Thomas M. Jordan[1], Michael A. Cooper[1], Dustin M. Schroeder[2], Christopher N. Williams[1], John D. Paden[3], Martin J. Siegert[4], and Jonathan L. Bamber[1]

[1]Bristol Glaciology Centre, School of Geographical Sciences, University of Bristol, Bristol, UK.
[2]Department of Geophysics, Stanford University, Stanford, California, USA.
[3]Center for Remote Sensing of Ice Sheets, University of Kansas, Lawrence, Kansas, USA.
[4]Grantham Institute and Department of Earth Science and Engineering, Imperial College, London, UK.

*Correspondence to:* T. M. Jordan (tom.jordan@bris.ac.uk).

**Abstract.** Subglacial roughness can be determined at variety of length scales from radio-echo sounding (RES) data; either via statistical analysis of along-track topography, or inferred from basal radar scattering. Past studies have demonstrated that subglacial terrain exhibits self-affine (fractal) scaling behaviour, where vertical roughness has a power-law relationship with the horizontal length scale.

5   A self-affine statistical framework, which enables a consistent integration of topographic roughness and radar scattering, has yet to be applied to RES. Here we do this for recent RES data from northern Greenland, and demonstrate that subglacial topography exhibits pronounced spatial variation in the Hurst (roughness power-law) exponent. A radar scattering model then enables us to explain how the Hurst exponent exerts strong topographic control upon radar scattering, which we map using the

10  waveform abruptness (pulse peakiness) parameter. Notably, lower abruptness (associated with diffuse scattering) occurs for regions with a higher Hurst exponent, and higher abruptness (associated with specular reflections) occurs for regions with a lower Hurst exponent. Finally, we compare the RES-derived data with an independent prediction for the subglacial thermal state of northern Greenland. This analysis shows that the majority of predicted thawed regions do not have the specular RES

15  scattering signature of deep subglacial lakes, and instead have a diffuse scattering signature.





## 1 Introduction

With the development of the newest generation of thermomechanical ice sheet models, there has been a growing awareness that better constraining the physical properties of the glacier bed is essential for improving their predictive capability (e.g. Price et al. (2011); Seroussi et al. (2013); Nowicki et al. (2013); Shannon et al. (2013); Sergienko et al. (2014); Ritz et al. (2015); Cornford et al. (2015)). Notably, the basal traction parameterisation - which encapsulates the thermal state, basal roughness, and lithology - is potentially the largest single geophysical uncertainty in projections of the response of ice sheets to climate change (Ritz et al., 2015). Distinction between frozen and thawed regions of the glacier bed is particularly important in constraining ice dynamics, since appreciable basal motion can only occur in regions where the glacier bed is wet (Seroussi et al., 2013; Macgregor et al., 2016). Airborne radio-echo sounding (RES) is the only existing remote sensing technique that can acquire bed data with sufficient spatial coverage to enable subglacial information to be obtained across the ice sheets (refer to Pritchard (2014) and Bamber et al. (2013a) for recent Antarctic and Greenland coverage maps). Often, however, there is great ambiguity in RES-derived subglacial information (Matsuoka, 2011), or RES-derived information is sub-optimal for directly applicability in ice sheet models (Wilkens et al., 2015). Subsequently, data analysis methods which seek to improve the clarity and glaciological utility of RES-derived subglacial information are undergoing a period of rapid development (e.g. Oswald and Gogineni (2008); Li et al. (2010); Fujita et al. (2012); Wolovick et al. (2013); Schroeder et al. (2013, 2016); Jordan et al. (2016)).

RES data-analysis methods for determining subglacial physical properties can be categorised in two ways; those which determine bulk properties (including the discrimination of basal water), and those which determine interfacial properties (subglacial roughness). Bulk material properties of the glacier bed can, in principle, be determined using the basal reflection coefficient (Bogorodsky et al., 1983; Peters et al., 2005; Jacobel et al., 2009; Schroeder et al., 2016). Performing basal reflection analysis on ice-sheet-wide scale is, however, greatly limited by uncertainty and spatial variation in englacial radar attenuation (Matsuoka, 2011; Matsuoka et al., 2012; Macgregor et al., 2012, 2015; Jordan et al., 2016). In contrast with bulk properties, subglacial roughness analysis methods are (near) independent of radar attenuation. Subglacial roughness can be determined either via statistical analysis of topography (typically spectral analysis) (Taylor et al., 2004; Siegert et al., 2005; Bingham and Siegert, 2009; Li et al., 2010; Rippin, 2013), or inferred from the electromagnetic scattering properties of the radar pulse (Oswald and Gogineni, 2008; Schroeder et al., 2014; Young et al., 2016)). Spectral analysis can provide valuable insight toward aspects past ice dynamics and landscape formation (Siegert et al., 2005; Bingham and Siegert, 2009; Rippin et al., 2014). However, since the technique is limited to investigating length scales greater than the horizontal resolution (typically $\sim 30$ m or greater), the relevance of the method in informing contemporary basal sliding physics - which requires metre-scale roughness information (Weertman, 1957; Nye, 1970; Hubbard et al., 2000; Fowler, 2011) - remains unclear. Radar scattering is sensitive to the length scale of the



electromagnetic wave (Shepard and Campbell, 1999) ($\sim$ 1-3 m in ice for the majority of airborne sounders), and can potentially reveal finer scale roughness information, including the geometry of

subglacial hydrological systems (Oswald and Gogineni, 2008; Schroeder et al., 2013, 2015; Young et al., 2016). High reflection specularity, such as occurs for deep (> 10 m) subglacial lakes (Oswald and Robin, 1973; Gorman and Siegert, 1999; Palmer et al., 2013), has been proposed to be a RES scattering signature that can aid in discrimination of thawed beds (Oswald and Gogineni, 2008, 2012).

Degrees of radar scattering can be mapped using either the waveform properties of the bed echo - e.g. the abruptness (pulse-peakiness) (Oswald and Gogineni, 2008), or by constraining the angular distribution of scattered energy - e.g. the specularity content (Schroeder et al., 2013; Young et al., 2016). Maps of both scattering parameters indicate defined spatial patterns, but, to date, have not been integrated with topographic-scale roughness analysis (horizontal length scales $\sim$ 10s of me-

tres and upwards). As such, there is a knowledge gap regarding the topographic control upon radar scattering. Observations indicate that subglacial roughness exhibits self-affine (fractal) scaling behaviour over length scales from $\sim 10^{-3}$ m to $\sim 10^{2}$ m (Hubbard et al., 2000; Macgregor et al., 2013). Self-affine scaling corresponds to when the vertical roughness increases at a fixed slower rate than the horizontal length scale, following a power-law relationship that is parameterised by the Hurst

exponent (Malinverno, 1990; Shepard et al., 2001). It is observed for a wide variety of natural terrain (Smith, 2014), including: the surface of Mars (Orosei et al., 2003); volcanic lava (Morris et al., 2008); and alluvial channels (Robert, 1988). If widely present, the self-affinity of subglacial roughness poses a challenge for integrating topographic roughness with existing glacial radar scattering models (Berry, 1973; Peters et al., 2005; Macgregor et al., 2013; Schroeder et al., 2015). This is be-

cause these statistically stationary models assume that roughness is independent of horizontal length scale, and an artificial scale-separation between high frequency roughness and low frequency topography is present (Berry, 1973). Radar scattering models with non-stationary, self-affine statistics naturally incorporate the multi-scale dependence of roughness, and are in widespread use in other fields of radar geophysics (e.g. Shepard and Campbell (1999); Franceschetti et al. (1999); Campbell

and Shepard (2003); Oleschko et al. (2003)).

In this study, we explore the connection between self-affine subglacial roughness and radar scattering for recently collected Operation Ice Bridge (OIB) RES data from the north western Greenland Ice Sheet (GrIS). We parameterise basal radar scattering using the waveform abruptness, which was previously incorporated in a basal thaw discrimination algorithm (Oswald and Gogineni, 2008,

2012). In their algorithm, high abruptness (specular reflections) is imposed as a necessary criteria for discrimination of basal thaw, and re-examining this algorithm in the context of self-affine statistics is a central goal. Firstly, we use examples from glacial topography to illustrate self-affine scaling and the role of the Hurst exponent (Sect. 2). We then describe along-track windowing methods that enable the spatial variation in topographic roughness and the Hurst exponent to be quantified from RES





data, along with the extraction of the waveform abruptness from the bed echo (Sect. 3). A self-affine radar scattering model, adapted from planetary radar (Shepard and Campbell, 1999; Campbell and Shepard, 2003), is then used to predict the relationship between the Hurst exponent and waveform abruptness (Sect. 4). The radar scattering model predicts that regions with higher Hurst exponent are associated with lower abruptness (diffuse scattering) and regions with lower Hurst exponent are associated with higher abruptness (specular reflections), which enables us to explain the spatial relationship between the Hurst exponent and waveform abruptness (Sect. 5.1, Sect. 5.2). Finally, we analyse the statistics of our RES-derived data in predicted thawed and frozen regions of the glacier bed (Sect. 5.3), using the recent basal thermal state synthesis mask in Macgregor et al. (2016). This analysis demonstrates that the RES scattering signature in thawed regions is, overall, more diffuse than in frozen regions, and indicates that the algorithm in Oswald and Gogineni (2008, 2012) is likely to yield both false-positive and false-negative discrimination of basal thaw.

## 2 Self-affine subglacial roughness

### 2.1 Overview

Statistical methods to extract the Hurst exponent, and thus to quantify self-affine scaling behaviour, are well established in the earth and planetary science literature (Malinverno, 1990; Shepard et al., 2001; Kulatilake et al., 1998; Orosei et al., 2003). These space-domain methods extract the Hurst exponent using the variogram (roughness verses profile length) and deviogram (roughness verses horizontal lag). Our motivation for use of these methods, rather than the spectral (frequency-domain) methods previously applied in studies of subglacial roughness (Taylor et al., 2004; Siegert et al., 2005; Bingham and Siegert, 2009; Li et al., 2010; Rippin, 2013)), is that they better reveal self-affine scaling behaviour (Turcotte, 1992; Shepard et al., 1995, 2001). Since the theory of self-affine roughness and related space-domain methods are not widely discussed in the glaciological literature - the only example being Macgregor et al. (2013) - we now provide a review of the key concepts. In order to illustrate the validity of the model to subglacial terrain, we use RES data from northern Greenland.

### 2.2 Interfacial roughness parameters

Topographic roughness can be measured by means of statistical parameters that are, in general, a function of horizontal length scale (Shepard et al., 2001; Smith, 2014). Two different interfacial roughness parameters - the root mean square (rms) height and rms deviation - are typically employed in self-affine roughness statistics (Shepard et al., 2001). The rms height is given by

$$\xi(L) = \left[ \frac{1}{N-1} \sum_{i=1}^{N} (z(x_i) - \bar{z})^2 \right]^{\frac{1}{2}}, \qquad (1)$$





where $N$ is the number of sample points within the profile window of length $L$, $z(x_i)$ is the bed elevation at point $x_i$, and $\bar{z}$ is the mean bed elevation of the profile. $\xi$ represents the standard deviation in bed elevation about a mean surface, and is a Gaussian-distributed random variable. The rms
deviation is given by

$$\nu(\Delta x) = \left[ \frac{1}{N} \sum_{i=1}^{N} [(z(x_i) - z(x_i + \Delta x)]^2 \right]^{\frac{1}{2}}, \tag{2}$$

where $\Delta x$ is the horizontal step size (lag). $\nu$ has a particular significance in the parameterisation of radar scattering models with self-affine statistics (Shepard and Campbell, 1999; Campbell and Shepard, 2003). We later demonstrate that $\nu$, rather than $\xi$, best enables a comparison to be drawn
between topographic scale roughness and radar scattering from glacial RES. The rms slope, which is proportional the rms deviation, is also widely used in self-affine statistics, but we do not do so here.

**2.3 Self-affine scaling behaviour and the role of the Hurst exponent**

Self-affine scaling (which is a sub-class of fractal scaling behaviour) can be parameterised using the Hurst exponent, $H$ (Malinverno, 1990; Shepard et al., 1995, 2001). $H$ quantifies the rate at which
roughness in the vertical direction increases relative to the horizontal length scale (and is defined for $0 \leq H \leq 1$). For a self-affine interface the following power-law relationships hold:

$$\xi(L) = \xi(L_0) \left( \frac{L}{L_0} \right)^H, \tag{3}$$

and

$$\nu(\Delta x) = \nu(\Delta x_0) \left( \frac{\Delta x}{\Delta x_0} \right)^H, \tag{4}$$

where where $L_0$ is a reference profile length, and $\Delta x_0$ is a reference horizontal lag (Shepard and Campbell, 1999; Shepard et al., 2001). Three limiting cases of self-affine scaling are typically discussed (Shepard and Campbell, 1999). Terrain with $H=1$ (where the roughness in the vertical direction increases at the same rate as the horizontal length scale) is referred to as 'self-similar'. Terrain with $H=0.5$, (where the roughness in the vertical direction increases with the square root of hor-
izontal length scale) is referred to as 'Brownian'. Terrain with $H=0$ (where the roughness in the vertical direction is independent of horizontal length scale) is referred to as 'stationary'. For a stationary ($H=0$) interface it follows from Eq. (3) and Eq. (4) that $\xi$ and $\nu$ are independent of $L$ and $\Delta x$ respectively.

We will later demonstrate that subglacial terrain exhibits near-ubiquitous self-affine scaling be-
haviour with pronounced spatial structure and variation for $H$. Four examples of 10 km along-track bed elevation profiles which encompass the range of observed $H$ values are shown in Fig. 1a, along with a zoom window to a 1 km track section in Fig. 1b. Clear differences are apparent between the different terrain examples. The black terrain and red terrain ($H \approx 0.9$ and $H \approx 0.7$) are between



Brownian and self-similar scaling, and exhibit 'persistent' trends where neighbouring points tend to
follow a general trend of increasing or decreasing elevation (Shepard and Campbell, 1999). A fea-
ture of terrain with these higher $H$ values is that it tends to appear rough at larger length scales (low
frequency) and smooth at smaller length scales (high frequency). By contrast, the green terrain ($H \approx$
0.3) is in the sub-Brownian scaling regime, and exhibits anti-persistent trends (where neighbouring
points tend to alternate between increasing and decreasing elevation). A feature of low $H$ terrain
such as this is that it tends to have similar roughness across length scales. The blue terrain ($H \approx 0.5$)
is close to an ideal Brownian surface, and exhibits no overall persistent trends. The 10 km profile
windows in Fig. 1a represent the length of along-track data over which we make our estimates for
$H$ (see Sect. 3.2).

**2.4 Calculation of $H$ via the variogram and deviogram**

In order to estimate $H$, and identify the length scale regime over which glacial terrain exhibits self-
affine behavior, $\xi$ and $\nu$ are plotted as a function of $L$ and $\Delta x$ respectively on double-logarithmic
scale plots; referred to as the variogram and deviogram respectively (Kulatilake et al., 1998; Shepard
et al., 2001). Variogram and deviogram plots for $\xi(L)$ and $\nu(\Delta x)$ for the four terrain examples in
Fig. 1 are shown in Fig. 2a and Fig. 2b respectively. It follows from Eq. (3) and Eq. (4) that, upon this
double-logarithmic scale, a straight line relationship is predicted for glacial terrain that is self-affine
with the gradient equal to $H$. In practice, a single self-affine relationship only holds over a limited
scale regime and a 'break-point' transition is often observed (Shepard et al., 2001). We describe how
we asses the break points for glacial terrain in Sect. 3.2, along with further details regarding the
application of the variogram and deviogram to along-track RES data. Fig. 2 clearly demonstrates the
significance of the Hurst exponent when assessing the relative roughness at different length scales,
with the green terrain ($H \approx 0.3$) becoming rougher relative to the other terrain at the smaller scales.

The space-domain variogram and deviogram have an (approximate) correspondence to the frequency-
domain power spectrum (Turcotte, 1992; Shepard et al., 1995, 2001). In frequency-space, self-affine
scaling occurs when the power spectrum, $S$, has a relationship of the form $S(k) \propto k^{-\beta}$ where $k$ is
the spatial frequency and $-\beta$ is the spectral slope. The relationship between $\beta$ and $H$ is dimension-
ally dependent, and for along-track data is given by $H = \frac{1}{2}(\beta - 1)$ (Turcotte, 1992). Despite this
correspondence, the space-domain methods are recommended to estimate $H$ as they are less noisy
and less likely to bias slope estimates than the power spectrum method (Shepard et al., 1995). The
study by Hubbard et al. (2000) observed self-affine scaling in the roughness power spectrum over
length scales from $\sim 10^{-3}$m to $\sim 10$ m for different sites across recently deglaciated terrain in the
immediate foreground of Glacier de Tsanfleuron, Switzerland. Their range for measured values of $\beta$
corresponds to $2.27 < \beta < 2.48$, which implies $H \sim 0.7$.





## 3   Analysis of RES data

### 3.1   Ice penetrating radar system and data coverage

The airborne RES data used in this study were collected by the Center for Remote Sensing of Ice
Sheets (CReSIS) within the Operation IceBridge (OIB) project, over the months March-May in
years 2011 and 2014. For all measurements the radar instrument, the Multichannel Coherent Radar
Depth Sounder (MCoRDS), was installed upon a NASA P-3B Orion aircraft. The sounder has a
frequency range from 180 to 210 MHz, corresponding to a centre wavelength $\sim 0.87$ m in ice.
After accounting for pulse shaping and windowing, this results in a depth-range resolution in ice of
$\sim 4.3$ m (Rodriguez-Morales et al., 2014; Paden, 2015). For the flight lines considered, the along-
track resolution after synthetic aperture radar (SAR) processing and multi-looking is $\sim 30$ m with an
along-track-sample spacing of $\sim 15$ m (Gogineni et al., 2014). The 2011 and 2014 field seasons were
used since they have a higher along-track resolution than other recent field seasons. Measurements
from MCoRDS are supplied as data products with different levels of additional processing (Paden,
2015). Level 2 data corresponds to ice thickness, ice surface, and bed elevation data, and is used to
calculate topographic-scale roughness and the Hurst exponent (Sect. 3.2). Level 1B data corresponds
to radar echo strength profiles, and is used to extract the waveform abruptness parameter from the
bed echo (Sect. 3.3). Basal reflection values can also be extracted from Level 1B data. We do not do
this here, since we do not wish to bias our interpretation due to uncertainty in radar attenuation.

The study focused upon data from north western Greenland, which encompasses measurements
close to three deep ice cores: Camp Century, NEEM and NorthGRIP (Fig. 3). The coverage map
is underlain by the predicted thermal state mask in Macgregor et al. (2016) (Fig. 11 in the original
study), which we later use to perform statistical analysis of basal RES data in different thermal
regions. This mask represents an up-to-date best estimate of thawed and frozen regions of the GrIS at
a 5 km grid scale, and is based upon a trinary classification: likely thawed (red), likely frozen (blue),
uncertain (grey). The mask was determined using four independent methods: thermomechanical
modeling of basal temperature, basal melting inferred from radiostratigraphy, modeling of surface
velocity, and surface texture observation. The mask is therefore independent of basal RES data, and
hence our RES-derived data fields. Our reason for selection of this region is threefold: firstly, the
region is similar to the RES basal thaw analysis by Oswald and Gogineni (2008, 2012); secondly,
confidence regarding the basal thermal state is high in the vicinity of the ice cores; and thirdly, the
data coverage for the 2011 and 2014 field seasons is of high density with a mixture of tracks in
predicted thawed and frozen regions.

### 220   3.2   Determination of topographic roughness and Hurst exponent from Level 2 data

The along-track spacing interval ($\sim 15$ m) of the Level 2 data is half the horizontal resolution ($\sim$
30 m), and the horizontal resolution represents the spacing at which bed elevation measurements



are considered as independent. Therefore, to remove local correlation bias, the Level 2 data were down-sampled, considering every second data point (corresponding to a $\sim 30$ m along-track spacing). Each flight track was then divided into 10 km along-track profile windows, as shown in the examples in Fig. 1a. Each profile window was then linearly detrended, which acts to reduce bias from a greater sampling of higher frequency than low frequency signal components (Shepard et al., 2001). The windows overlap with a sample spacing of 1 km, with the centre of each window defined to be the point at which $H$ and the roughness parameters are geolocated to. This 'moving window' approach was employed as it enables greater continuity in the estimates for $H$. Prior to estimating $H$, $\xi(L)$ and $\nu(\Delta x)$ were computed following Eq. (1) and Eq. (2) respectively. These calculations used the 'interleaving' sampling method described in (Shepard et al., 2001), which enables all of the data points to be sampled effectively. Our windowing method is similar to that described in Orosei et al. (2003) for the self-affine charaterisation of Martian topography, where a non-overlapping 30 km window was assumed. Our choice of 10 km for the profile window and 1 km for the effective resolution, represent a good trade-off between resolution and the smoothness of the derived data fields.

In this study we are interested in computing $H$ at the length scale of the Fresnel zone ($\sim 100$ m), since this enables the most accurate parameterisation of the radar scattering model described in Sect. 4. The focus on smaller length scales is also is consistent with the recommendations in Shepard et al. (2001), due to the break point transitions that can occur in self-affine scaling behaviour. For the data we consider, the lower bounds of the horizontal length scales are $\sim 90$ m for $\xi(L)$ (since three elevation measurements are the minimum required to calculate $\xi(L)$ using Eq. (1)) and $\sim 30$ m for $\nu(\Delta x)$. $\nu(\Delta x)$ therefore better enables the estimation of $H$ at smaller length scales and we focused upon the deviogram method, Fig. 2b, (although we also used the variogram, Fig. 2a, for comparative purposes). Additionally, as suggested in Fig. 2 the relationships for $\nu(\Delta x)$ are, in general, significantly smoother than $\xi(L)$. The upper length scales in the deviogram and variogram were set to be $\Delta x$=1 km and $L$=1 km respectively, which follows from the recommendation by Shepard et al. (2001) that at least 10 independent sections of track are used in the calculations. As shown in Fig. 2, the gradients ($H$) were calculated using the first five data points (which, for the deviogram, is over the range $\Delta x \sim 30$ m to $\Delta x \sim 150$ m). Self-affine scaling behavior often extends beyond these length-scales and we estimated the break points for $\xi(L)$ and $\nu(\Delta x)$ using a segmented linear regression procedure. Briefly, this involved firstly calculating the gradient ($H$) for the first five data points. Additional data points at increasing length scales were then added into each linear regression model, and the gradient was recalculated. A stopping criteria was then applied where if the new gradient was found to exceed a specified tolerance from the original estimate, then a break point was identified.



### 3.3 Determination of waveform abruptness from Level 1B data

The processing of the Level 1B data (the basal waveform) uses the procedure described in Jordan
et al. (2016), which, in turn, is largely based upon Oswald and Gogineni (2008). Firstly, this in-
volved performing an along-track average of the basal waveform, where adjacent basal waveforms
are stacked about their peak power values and arithmetically averaged. This averaging approach is
phase-incoherent and acts to smooth power fluctuations due to electromagnetic interference (Os-
wald and Gogineni, 2008). The size of the averaging window varies as a function of Fresnel zone
radius, and subsequently each along-track averaged waveform to corresponds to approximately a
separately illuminated region of the glacier bed (see Jordan et al. (2016) for details). The degree of
radar scattering is quantified using the waveform abruptness

$$A = \frac{P_{peak}}{P_{agg}}, \tag{5}$$

where $P_{peak}$ is the peak power of the bed echo and $P_{agg}$ is the aggregated power, which is calculated
by a discrete summation of the bed echo power measurements in each depth range bin. $P_{agg}$ was
introduced by Oswald and Gogineni (2008) since, based upon energy conservation arguments, it is
argued to be more directly related to the predicted (specular) reflection coefficients than equivalent
peak power values. In radar altimetry, the waveform abruptness is commonly called 'pulse peakiness'
(e.g. Peacock and Laxon (2004); Zygmuntowska et al. (2013)).

Fig. 4 shows three examples of basal waveforms, along with their corresponding $A$ values. Ob-
served values of $A$ range from ∼ 0.03 to 0.60, and in Sect. 4.3 we theoretically constrain the max-
imum value to be 0.65. Higher $A$ values are associated with specular reflections from smoother
regions of the glacier bed, whilst lower $A$ values are associated with diffuse reflections from rougher
regions (Oswald and Gogineni, 2008). When calculating the summation for $P_{agg}$, a signal-noise-
ratio threshold was implemented by testing for decay of the peak power to specified percentage
above the noise floor. Thresholds of 1, 2, and 5 % were considered and 2 % was found to give
the best coverage, whilst excluding obvious anomalies. Due to this quality filtering step there are
therefore sometimes small gaps in the along-track $A$ data.

The basal waveform (and hence the calculated values of $A$), results from a superposition of along-
track and cross track energy (Young et al., 2016). Subsequently, the anisotropy of radar scattering
(and inferences regarding the anisotropy of subglacial roughness), is not explicitly revealed by $A$.
Hence, the studies of Oswald and Gogineni (2008, 2012) treat $A$ as an isotropic parameter, and we
follow this approach here.



## 4  Radar scattering model for self-affine roughness

### 4.1  Overview

The waveform abruptness has previously been discussed without reference to roughness statistics, and here we do this using a self-affine radar scattering model. Radar scattering models from natural terrain fall into two different categories: 'coherent' which incorporate deterministic phase interference and 'incoherent' which incorporate random phase interference (Ulaby et al., 1982; Campbell and Shepard, 2003; Grima et al., 2014). Coherent scattering models are applicable where the reflecting region is orientated near-perpendicular to the incident pulse (the nadir regime) and the reflecting region is fairly smooth at the scale of the illuminating wavelength (Campbell and Shepard, 2003), which is normally assumed be a good approximation for the RES of glacier beds (Peters et al., 2005; Macgregor et al., 2013; Schroeder et al., 2015).

Below we describe and adapt a coherent scattering model, first developed for the nadir regime of planetary radar sounding measurements, which incorporates self-affine roughness statistics (Shepard and Campbell, 1999; Campbell and Shepard, 2003). The model is parameterised using the Hurst exponent values derived from the subglacial topography (Sect. 3.2), and thus enables a connection to be made between the topographic roughness and radar scattering. Coherent scattering models can be used to model decrease in specularly reflected power as a function of rms roughness (Berry, 1973; Peters et al., 2005), and this is the central aspect of the model which we focus upon here. Specifically, we show that, under assumptions of energy conservation, this power decrease can be used to predict the relationship between the Hurst exponent and waveform abruptness.

### 4.2  Modeling the coherent power

The physical assumptions behind the self-affine scattering model are summarised in Shepard and Campbell (1999). The central assumption that differentiates the model from coherent stationary ($H{=}0$) models (Berry, 1973; Peters et al., 2005; Macgregor et al., 2013; Grima et al., 2014; Schroeder et al., 2015), is that the rms height increases as a function of radius, $r$, about any given point, following the self-affine relationship

$$\xi(r) = \frac{1}{\sqrt{2}}\nu_\lambda \left(\frac{r}{\lambda}\right)^H ,\tag{6}$$

where $\nu_\lambda = \nu(\Delta x = \lambda)$ is the wavelength scale rms deviation. Equation (6) assumes radial isotropy for $H$ and $\xi$, and, since we are focusing upon constraining the (near) isotropic abruptness parameter, is a justifiable approximation. The statistical distribution for $\xi(r)$ is assumed to be Gaussian, which is similar to most $H{=}0$ models (but with an additional radial dependence.) Via $\nu_\lambda$, the self-affine model is explicitly formulated with respect to the horizontal scale of rms roughness. The radio wavelength of MCoRDS in ice is $\sim 0.87$ m (Paden, 2015), and hence wavelength scale rms deviation is approximately equivalent to metre scale rms deviation. An unavoidable caveat to the parameterisation of the





radar scattering model using Eq. (6) is that the $H$ values derived from the topography (length scale $\sim$ 30-150 m) are extrapolated downwards to the wavelength scale.

An expression for the radar backscatter coefficient (radar cross-section per unit area) is then derived by considering a phase variation, $\frac{4\pi\xi(r)}{\lambda}$, integrated across the Fresnel zone (Shepard and Campbell, 1999; Campbell and Shepard, 2003). For nadir reflection the radar backscatter coefficient is given by

$$\sigma_0 = \frac{16\pi^2 R_e^2}{\hat{r}_{max}^2} \left( \int_0^{\hat{r}_{max}} \exp\left[ -\frac{4\pi^2}{\lambda^2} \nu_\lambda^2 \hat{r}^{2H} \right] \hat{r} d\hat{r} \right)^2 , \qquad (7)$$

where $\hat{r} = \frac{r}{\lambda}$ is the wavelength-scaled radius, $\hat{r}_{max}$ is the wavelength-scaled radius of the illuminated area (the Fresnel zone), and $R_e$ is the reflection coefficient for the electric field (Campbell and Shepard, 2003). The coherent power, $P$, can then obtained by dividing Eq. (7) by $4\pi^2 \hat{r}_{max}^2$ (a geometric factor which follows from the backscatter coefficient of a flat conducting plate (Ulaby et al., 1982)) to obtain

$$P = \frac{4R_e^2}{\hat{r}_{max}^4} \left( \int_0^{\hat{r}_{max}} \exp\left[ -\frac{4\pi^2}{\lambda^2} \nu_\lambda^2 \hat{r}^{2H} \right] \hat{r} d\hat{r} \right)^2 . \qquad (8)$$

For the case where $H$=0, $\xi(r)$ in Eq. (6) is independent of radius. It follows that $\xi^2 = \frac{1}{2}\nu_\lambda^2$ and the exponent in Eq. (8) is also independent of radius which gives

$$P = R_e^2 \exp\left( -\frac{16\pi^2}{\lambda^2} \xi^2 \right). \qquad (9)$$

Equation (9) is the same power decay formula as coherent $H$=0 models (Peters et al., 2005; Mac-
gregor et al., 2013; Grima et al., 2014; Schroeder et al., 2015), where it is sometimes multiplied by a first order Bessel function (which enables some of the incoherent energy contribution to be captured (Macgregor et al., 2013)). Thus the stationary limit of the model that we use is consistent with previous glacial basal scattering models. It is clear that the coherent power for the self-affine model, Eq. (8), has two roughness degrees of freedom: $H$ and $\nu_\lambda$, which can be conceptually related to the
gradient and the intercept of the deviogram (Fig. 2). This contrasts with the stationary model, Eq. (9), which has one degree of freedom: $\xi$.

### 4.3   Relationship between the Hurst exponent and waveform abruptness

The utility of the waveform abruptness in quantifying different degrees of scattering, rests upon the assumption that the majority of the overall energy is contained within the echo envelope (Oswald and
Gogineni, 2008). In other words, it is assumed that, for reflection from the same bulk material, the aggregated/integrated power from a rough interface ($\nu_\lambda > 0$) is equivalent to the peak power from a given smooth interface: i.e. $P_{agg} \approx P(\nu_\lambda = 0)$. This energy equivalence was demonstrated to hold well for the waveform processing procedure and Greenland RES systems by Oswald and Gogineni





(2008). It follows from this energy equivalence that the abruptness, $A$, can be expressed in terms of

the coherent power, Eq. (8), as

$$A = \frac{P_{peak}}{P_{agg}} = C \frac{P(\nu_\lambda)}{P(\nu_\lambda = 0)},$$

(10)

where $C$ is a proportionality constant that corresponds to the theoretical maximum abruptness value which occurs when the radar pulse is specularly reflected and $P_{agg} = P_{peak}$. For specular reflection the pulse is the shape of compressed chirp with the pulse width determined by the signal bandwidth,

and $C$ can be estimated from the number of range cells which fit in the depth-range resolution ($\sim$ 0.65). Finally, substituting Eq. (8) into Eq. (10) gives

$$A = \frac{4C}{\hat{r}_{max}^4} \left( \int_0^{\hat{r}_{max}} \exp\left[ -\frac{4\pi^2}{\lambda^2} \nu_\lambda^2 \hat{r}^{2H} \right] \hat{r} d\hat{r} \right)^2 .$$

(11)

As is the case for $P$ in Eq. (8), Eq. (11) has two roughness degrees of freedom: $H$ and $\nu_\lambda$. Shepard and Campbell (1999) note that the primary dependence for $P$, (and hence $A$), is upon $H$; with a

weaker secondary dependence upon $\nu_\lambda$. In order to illustrate this dependency, we firstly consider the relationship between $A$ and $H$ for fixed $\nu_\lambda$ (Fig. 5a), and secondly the relationship between $A$ and $\nu_\lambda$ for fixed $H$ (Fig. 5b). Fig. 5a demonstrates that higher values of $\nu_\lambda$ (the black curve), result in negligible $A$ for all but the lowest values of $H$. Intermediate values of $\nu_\lambda$ (the red and blue curves), exhibit a sharp transition from higher to lower values of $A$ as $H$ increases. Low $\nu_\lambda$ (the green curve)

has high $A$ for all $H$. Fig. 5b demonstrates a monotonic decrease in $A$ with $\nu_\lambda$ for each value of $H$, with the decay length decreasing rapidly with increasing $H$.

The physical explanation for the strong dependence of the coherent power upon $H$, and the relationships which we observe in Fig. 5, is discussed by Shepard and Campbell (1999) and Campbell and Shepard (2003). It relates to the fact that significant coherent returns can only occur from annu-

lar regions where $\xi(r) < \left(\frac{\lambda}{8}\right)$ (the Rayleigh criterion). It follows from Eq. (6) that high values of $H$ lead to a rapid increase in roughness with radius that rapidly exceeds this threshold. Subsequently, for high $H$ interfaces, the roughness at the wavelength scale, $\nu_\lambda$, must be a couple orders of magnitude smaller than the Rayleigh criterion to enable significant coherent returns (i.e. non-negligible $A$). The curves in Fig. 5 assume $\hat{r}_{max}=100$ (corresponding to a Fresnel zone radius $\sim 115$ m for

the ice wavelength $\sim 0.87$ m). In general, the relationships in Fig. 5 are insensitive to this choice of radius. This is because the radii of the coherent annular regions are typically significantly less than the Fresnel zone, and thus act as the dominant length scale for the integration limit in Eq. (11).

## 5   Results

Firstly, we describe maps for the Hurst exponent, rms deviation (topographic roughness), and the

waveform abruptness (radar scattering) in northern Greenland (Sect. 5.1). Secondly, we compare the observed relationship between the Hurst exponent and waveform abruptness with the predictions of




the self-affine radar scattering model (Sect. 5.2). Thirdly, we perform a statistical analysis of the Hurst exponent and the waveform abruptness in predicted thawed and frozen regions of the glacier bed corresponding to the predicted thermal state mask in Macgregor et al. (2016) (Sect. 5.3). Since

our primary focus is exploring the relationship between the Hurst exponent and waveform abruptness (which we demonstrate to be near isotropic), we choose not to investigate roughness anisotropy in detail.

### 5.1   Maps for Hurst exponent, topographic-scale roughness, and waveform abruptness

A map for $H$, underlaid by the frozen-thawed mask, for the north western GrIS is shown in Fig.

6a. The calculations use the deviogram for $\nu(\Delta x)$, and $H$ is estimated over the horizontal length scale $\Delta x \sim$ 30-150 m. In general, higher values of $H$ occur toward the margins of the ice sheet (generally, but not exclusively, thawed regions) and lower values toward the interior (generally, but not exclusively, frozen regions). The mean $r^2$ value for the deviogram linear regression fits was $\sim 0.99$, indicating a very high overall correlation of the linear model. These correlation fits are,

however, probably artificially high due to the smoothening of the data that occurs when applying the windowing and repeated sampling approach described in Sect. 3.2, and are unlikely to represent the true accuracy of the RES-derived data. There are clearly some discontinuities in the along-track maps in 6a which are likely explained by either anisotropy in $H$ (due to underlying roughness anisotropy); or due the self-affine terrain model breaking down in certain regions (e.g. a sharp terrain discontinuity

such as a subglacial cliff).

Calculation of $H$ was also performed using the variogram for $\xi(L)$, over the length scale $L \sim 90$m to $L \sim 210$ m. The map for $H(\xi(L))$ (not shown) has the same overall spatial distribution as in Fig. 6a but with greater high frequency noise apparent and a correspondingly lower mean $r^2$ value ($\sim$ 0.96). Due to the differing length scales in the $H(\xi(L))$ and $H(\nu(\Delta x))$ estimates, direct cross-over

analysis is not possible. However, differencing the estimates as $H(\nu(\Delta x))$-$H(\xi(L))$, gives a small mean bias of -0.026 and a cross-over standard deviation of 0.10 (10% of the parameter range). This is likely to better represent the accuracy of the data than the linear model fit uncertainties. Repeat cross-over analysis using different profile window sizes for the processing in Sect. 3.2 (e.g. 15 km) confirms that 0.10 serves a reasonable estimate for uncertainty of $H$.

Maps for $\nu(\Delta x$=30 m) (the horizontal resolution and the smallest length scale in the deviogram calculation for $H$), $\nu(\Delta x$=150 m) (slightly larger than the typical size of the Fresnel zone, and the largest length scale used in the deviogram) are shown in Fig. 6b and c respectively. The spatial distributions indicate that, at each respective length scale, relatively rough terrain is present toward the margins of the ice sheet (typically, but not exclusively, thawed regions), and relatively smooth

terrain is present in the interior (typically, but not exclusively, frozen regions). Despite the clear spatial variation in $H$ in Fig. 6a, the overall spatial distribution for $\nu(\Delta x$=30 m) and $\nu(\Delta x$=150 m) are remarkably similar. Thus, from a purely visual inspection of the topographic-scale roughness,





the pronounced spatial variation in self-affine scaling behaviour is not apparent. As with the map
for $H$, the rms deviation data at some of the flight track intersections is suggestive that a degree of
roughness anisotropy is present.

A map for $A$ is shown Fig. 6d. There is a high degree of spatial structure, with near-continuous
regions of higher $A$ (specular reflections) present in parts of the frozen interior, and also toward
the north western margin near Camp Century. Values of $A$ range from $\sim 0.03$ to $0.60$ which is in
agreement with our theoretically constrained maximum abruptness value of 0.65 determined in Sect.
4. Cross-over analysis from flight track intersections indicates that variation in $A$ is $\sim 0.05$ (less
than $10\%$ the parameter range). This variation is approximately the same for inter- and intra- field
campaign cross-over analysis (although this is to be expected since the radar systems are similar).
This variation also incorporates any minor variation due to anisotropy, and confirms that $A$ is near-
isotropic. Similar continuous regions of high abruptness in the northern interior were observed by
Oswald and Gogineni (2008, 2012).

As part of the analysis we also considered estimation of the breakpoint transitions for $\xi(L)$ and
$\nu(\Delta x)$ using the segmented linear regression procedure described Sect. 3.2. The exact values of the
breakpoints depend upon how strict the stopping criteria is, so here we just discuss some general
trends. Firstly, the self-affine scaling relationships often extend over a much greater length scale
than the upper length scale used in the $H$ estimates (often over 500 m as occurs in Fig. 2). Secondly,
as is also the case in Fig. 2, the breakpoints for $\nu(\Delta x)$ generally occur at greater length scales than
for $\xi(L)$. Thirdly, the break points for both $\nu(\Delta x)$ and $\xi(L)$ tend to be greater toward the ice sheet
margins.

### 5.2    Observed relationship between Hurst exponent and waveform abruptness

The total frequency distribution for $H$, corresponding to the along-track calculations in Fig. 6a, is
shown in Fig. 7a. The distribution is divided into three categories: (i) $H > 0.75$ ('High' $H$); (ii)
$0.5 < H \leq 0.75$ ('Medium $H$'); (iii) $H \leq 0.5$ ('Low' $H$), which we later employ in comparative
analysis with the waveform abruptness. Approximately $0.1\%$ of the $H$ estimates are $> 1$ and none
of the $H$ estimates are $< 0$, representing near-ubiquitous self-affine scaling behaviour, ($0<H<1$). A
overall negative skew for the distribution of $H$ is observed with a mean value of 0.65, indicating
that the majority of the subglacial terrain along the flight tracks lies between Brownian ($H=0.5$)
and self-similar ($H=1$) scaling regimes. The spatial coverage of the radar flight tracks in Fig. 6a is,
however, more comprehensive in regions of higher $H$. Thus the mean value and skew of $H$ in Fig.
7a are likely overestimates and underestimates of true (equal area) averaged values for the region of
the northern GrIS in Fig. 3.

The self-affine coherent scattering model, Sect. 4, predicts that there are two roughness degrees of
freedom that control $A$: $H$ (the primary control) and $\nu_\lambda$ (the secondary control). Given the primary
dependence of $A$ upon $H$, a natural starting point is to compare the model predictions with the





observed relationship between $A$ and $H$. Based upon the assumption that $\nu_\lambda$ (which is not observed directly) varies spatially, a statistical distribution for $A$ is theoretically predicted for each value of $H$ which corresponds to the family of predicted curves in $H$-$A$ space, Fig. 5a. Additionally, since high $A$ is predicted to be suppressed for high $H$, we would expect there to be a statistically lower mean $A$ value for the High $H$ than Low $H$ category Fig. 7a. In other words; a statistically-distributed inverse relationship is predicted.

In order to test these predictions, we considered the statistics of the three separate $A$ distributions for each $H$ category, which are shown for: High $H$ in Fig. 7b, Medium $H$ in Fig. 7c, and Low $H$ in 7d. These categories correspond to approximately 30 %, 50 % and 20 % of the total data respectively. A nearest neighbour interpolation was used to pair each $A$ value ($\sim$ 100-150 m along-track spacing) with each $H$ value (1 km along-track spacing). The lowest mean value, smallest variance, and strongest positive skew is observed for the High $H$ category. This supports the general prediction in Fig. 5 that higher $A$ values (specular reflections) are suppressed in regions of higher $H$, with lower $A$ values (diffuse scattering) being more probable. The highest mean value, greatest variance, and weakest positive skew is observed for the Low $H$ category. This supports the prediction in Fig. 5 that $A$ is less constrained in regions of lower $H$, with a tendency toward higher values (specular reflections). As would be expected, the $A$-distribution statistics for the Medium $H$ category lie between the High $H$ and Low $H$ categories with intermediate mean values, variance, and skewness.

We repeated the above analysis for $H$ values that were estimated using the variogram for $\xi(L)$. The distributions for the three $H$ categories are qualitatively similar to Fig. 7b-d but with less pronounced quantitative differences between $H$ categories (e.g. the mean values for $A$ are 0.201, 0.243 and 0.254 for High, Medium and Low $H$ categories respectively). This is as expected, since the length scales at which $H$ is estimated using the deviogram are more directly related to radar scattering than the variogram.

### 5.3 Statistics in thawed and frozen regions

The spatial distributions in Fig. 6 indicate that predicted thawed regions of the northern GrIS (using the basal thermal state mask in Macgregor et al. (2016)) have: higher values of the Hurst exponent, higher values of rms deviation (higher topographic-scale roughness), and lower values of waveform abruptness (more diffuse scattering), than frozen regions. In order to quantify these observations, we considered statistical distributions for thawed/frozen subsets of the along-track data. Distributions for $H$ in thawed/frozen regions (corresponding Fig. 6a) are shown in Fig. 8a and b; distributions for $\nu(\Delta x=30$ m) in thawed/frozen regions (corresponding to Fig. 6b) are shown in Fig. 8c and d; and distributions for $A$ in thawed/frozen regions (corresponding to Fig. 6d) are shown in Fig. 8e and d.

The distributions for all RES-derived data exhibit pronounced statistical differences between thawed and frozen regions. The mean value of $H$ in thawed regions is 0.74 with a strong negative skew, whereas the mean value of $H$ in frozen regions is 0.54 with a weak negative skew. The



mean value of $\nu(\Delta x$=30 m) in thawed regions is 6.36 m, which is over double the mean value of 2.80 m in frozen regions. A qualitatively similar distinction between thawed and frozen regions is also present for $\nu(\Delta x$=150 m), with a mean value of 21.7 m in thawed regions and 7.2 m in frozen regions (not shown). The thawed distribution for $A$ is similar to the high $H$ category in Fig. 7a, with a mean $A$ value of 0.165 and strong positive skew. The frozen distribution is similar to the low $H$

category in Fig. 7a with a mean $A$ value of 0.264 and a weak positive skew. An anti-correlation between the skewness of the $H$ and $A$ distributions is also observed (i.e. in thawed regions a strong negative skew for $H$ is paired with a strong positive skew for $A$, and in frozen regions a weak negative skew for $H$ is paired with a weak positive skew for $A$). This provides further observational evidence for the applicability of the self-affine radar scattering model (Sect. 4), which predicts a

statistically-distributed inverse relationship between $H$ and $A$.

## 6   Discussion

Our study demonstrates that the Hurst exponent exerts strong topographic control upon the spatial distribution of the waveform abruptness (radar scattering). Notably, higher values of abruptness (specular reflections) are suppressed in regions of higher Hurst exponent, with lower abruptness

(diffuse scattering) being more typical. Additionally, extended continuous regions of higher abruptness are generally limited for lower Hurst exponent. Whilst clearly idealised - assuming both radial isotropy and a downward extrapolation of $H$ - the self-affine radar scattering model predicts this behaviour, and thus establishes a causative link between the spatial distribution of self-affine roughness and radar scattering. This finding implies that maps of radar scattering information - including

both the waveform abruptness parameter discussed in here and in Oswald and Gogineni (2008) and Oswald and Gogineni (2012), and the specularity content in Schroeder et al. (2013) and Young et al. (2016) - will, in future, benefit from analysis that takes into account self-affine topographic control.

The waveform abruptness parameter was previously employed by Oswald and Gogineni (2008, 2012), in the context of basal thaw discrimination. In their algorithm regions of thaw are discrimi-

nated if: (i) the relative reflection coefficient is above a threshold (with reflection defined using the aggregated power (see Sect. 3.3), and an attenuation model where the attenuation rate has an inverse relationship with surface elevation); (ii) the abruptness is also above a threshold. Thus, in their approach, high abruptness (specular reflections) is a necessary, but not sufficient, criteria for identifying basal thaw. Conceptually, their approach assumes that thawed regions have a similar RES

signature to deep (depth > 10 m) subglacial lakes which exhibit brighter and more specular reflections than surrounding regions (Oswald and Robin, 1973; Palmer et al., 2013). Shallower (depth < 10 m) subglacial lakes often produce more diffuse reflections, due to scattering from the lake bottom and related interference effects Gorman and Siegert (1999). The exact depth at which a lake is





'shallow' or 'deep', based upon its scattering signature, is dependent on the electrical conductivity,
and 10 m serves as a rough guide (refer to Gorman and Siegert (1999) for a full discussion).

Our statistical analysis of abruptness in predicted frozen and thawed regions (Sect. 5.3) demonstrates that, overall, very different RES scattering signatures are likely to be present than were assumed by Oswald and Gogineni (2008, 2012). Firstly, the majority of the predicted thawed regions have lower abruptness (associated with diffuse scattering). In their algorithm, this would correspond to false-negative detection of basal thaw (since the necessary high abruptness condition for thaw is not satisfied). Secondly, high abruptness is often present in predicted frozen regions, many of which are interpreted as thawed by Oswald and Gogineni (2008, 2012) (e.g. parts of the region of near-continuous higher abruptness around the Camp Century ice core). Since the original work of Oswald and Gogineni (2008), the role that uncertainty in radar attenuation has in biasing the spatial distribution of radar reflection has become much better understood (Matsuoka, 2011; Macgregor et al., 2012; Jordan et al., 2016). For example, if an attenuation model has a constant systematic bias in attenuation rate, then there will be a thickness-correlated bias in estimated basal reflection values (Jordan et al., 2016). Thus, it is possible that regions predicted to have high values of relative reflection in the frozen interior (Oswald and Gogineni, 2008, 2012) do so due to spatially correlated bias in the attenuation model. In turn, combined with high abruptness, this would lead to a false-positive identification of basal thaw.

It is important to note that both the predictions of the self-affine radar scattering model (Sect. 4), and the observed relationship between the abruptness and Hurst exponent (Sect. 5.2), are consistent with the specular RES scattering signature that we would expect from electrically deep subglacial lakes. Under the self-affine roughness framework, a large geometrically flat feature such as a lake would have a negligible value of $H$ and $\nu_\lambda$. This scenario occurs for the low $H$ limit of the green curve in Fig. 5a, where predicted values for $A$ are $\sim 0.65$ (corresponding to a perfectly specular reflection). Whilst the majority of the thawed regions have lower abruptness, there are some smaller, localised, patches of higher abruptness present in Fig. 6d: for example, in the main trunk of Petermann glacier (80.5° N, 59.5° W), and the flight lines just to the North and East of NorthGRIP. Whilst these regions are consistent with there being deep water present (in the sense that specular reflections are observed in a region predicted to be above pressure melting point), on the basis of the RES scattering signature alone it is not possible to confirm that this is the case. This is because the frozen abruptness distribution in Fig. 8f indicates that basal water is not required to produce highly specular reflections, and thus smooth regions of bedrock may be responsible for the high abruptness. The presence of at least some localised patches of high abruptness in thawed regions is also consistent with the recent discovery of two small subglacial lakes in north western Greenland of $\sim$ 8 km$^2$ and $\sim$ 10 km$^2$ in extent (Palmer et al., 2013). More generally, however, the relative rarity of high abruptness in thawed regions is in agreement with a recent hydrological potential analysis



(Livingstone et al., 2013), which predicted that deep subglacial lakes are both rare and small in our
region of interest.

The Hurst exponent provides information about the relationship that exists between vertical rough-
ness and the horizontal length scale. Whilst it is related to the slope of the roughness power spectrum,
past spectral analysis of glaciological terrain tends to obscure this information (since an integrated

'total roughness' metric is typically used) (Taylor et al., 2004; Siegert et al., 2005; Bingham and
Siegert, 2009; Li et al., 2010; Rippin, 2013). Subsequently, the Hurst exponent represents new sub-
glacial roughness information, that could potentially be utilised much more widely than our current
application in constraining radar scattering. For example, planetary scientists have previous em-
ployed the Hurst exponent in a geostatistical classification of Martian terrain (Orosei et al., 2003).

Interestingly, the spatial distribution of the Hurst exponent for the Martian surface, has a similar
level of spatial variation and coherence to what we observe for glacial terrain. Additionally, the dis-
tribution of $H$ for Martian terrain is skewed toward higher, self-similar, values with near-continuous
regions of lower $H$ limited to mid-latitude plains. For Greenland, this self-affine statistical landscape
classification could be integrated with existing knowledge of geology (e.g. Henriksen (2008)) and

preglacial landscape features; including paleofluvial drainage networks (Cooper et al., 2016) and
paleofluvial canyons (Bamber et al., 2013b).

Whilst topographic (spectral) roughness analysis has already been performed for topography be-
neath ice sheets (e.g. for the Greenland Ice Sheet by Rippin (2013), and the East Antarctica Ice Sheet
by Siegert et al. (2005)), analysis is limited to a horizontal length scale that is greater than the along-

track resolution (typically $\sim 30$ m or greater). This minimum observable length scale is significantly
greater than the dominant (metre) length scale at which the processes that modulate basal sliding -
enhanced plastic flow and relegation - operate at (Weertman, 1957; Nye, 1970; Hubbard et al., 2000;
Fowler, 2011). The maps for topographic-scale rms deviation in Fig. 6b and c, and related analysis
in Sect. 5.3, both demonstrate that rougher terrain at the topographic scale occurs in thawed regions

rather than frozen regions. This is the opposite behaviour to what we would expect if the roughness
measure plays a direct role in modulating basal sliding, confirming the view that the length-scale is
likely to be too large to be relevant. Since it is formulated with respect to wavelength (approximately
metre) scale roughness, the self-affine radar scattering model, provides a way to estimate metre-scale
roughness (i.e. given $A$ and $H$ obtain an estimate for $\nu_\lambda$ in accordance with the curves in Fig. 5).

There is an obvious caveat, however, that the basal roughness parameters used in this study - rms
height and rms deviation - are chosen because they enable the parameterisation of radar scattering
(not for their direct relevance to basal sliding physics). Therefore, as is being explored in other recent
subglacial roughness works (Li et al., 2010; Rippin et al., 2014; Wilkens et al., 2015), these radar
roughness parameters could potentially be modified for glaciological application.

The Hurst exponent has previously been shown to play a dynamical role in the flow resistance of
alluvial channels (Robert, 1988). Whilst basal sliding is clearly a different physical phenomena, it is



possible that the Hurst exponent itself may provide a useful radar-derived parameter for informing
our understanding of geometric control upon this process. In Sect. 5.1 and Sect. 5.3 we observed
that predicted thawed regions of the glacial bed are characterised by higher (often near self-similar)

values of the Hurst exponent, whereas frozen regions are characterised by lower values (normally
Brownian, or sub-Brownian). One could therefore conjecture that the persistent behaviour (neigh-
bouring points follow similar elevation trend) associated with high $H$ interfaces (see Sect. 2.3) could
act to promote basal sliding, whereas the anti-persistent behaviour (neighboring points alternate in
elevation trend) associated with low $H$ interfaces could act to inhibit basal sliding. However, as is

widely acknowledged, attributing a direct link between subglacial roughness and contemporary ice
dynamics is a complex topic (Siegert et al., 2005; Bingham and Siegert, 2009; Rippin et al., 2014).
Therefore, as with other measures or basal roughness, the spatial variation in the Hurst exponent
is likely to both originate from, and influence, different glaciological processes at a variety of spa-
tial scales. Additionally, we recommend that future works which investigate the connection between

the Hurst exponent and glaciological processes should be discussed with reference to anisotropy (in
particular with respect to the flow direction).

The pronounced spatial heterogeneity of $H$ implies that estimation of roughness statistics from
$H$=0 radar scattering models (Eq. (9) and (Berry, 1973; Ulaby et al., 1982; Peters et al., 2005; Grima
et al., 2014)) may give erroneous results; particularly when comparing the overall spatial distribution

between regions with different $H$ values. We did not consider the anisotropy of the Hurst exponent
in our radar scattering model, which was justifiable because we were interested in understanding
how the Hurst exponent relates to the (near) isotropic abruptness. However, in certain regions of
the ice sheets, basal radar scattering is known to be highly anisotropic; as revealed by maps of the
specularity content for Thwaites glacier by Schroeder et al. (2013) and Byrd Glacier by Young et al.

(2016). Thus a clear direction of future research would be to modify the self-affine radar scattering
model (Sect. 4) to take into account anisotropy in $H$, and then to compare this model with maps for
the specularity content.

Finally, geostatistically-based interpolation methods which employ aspects of self-affine statistics
(Goff and Jordan, 1988), have found recent application in generating synthetic subglacial topography

(Goff et al., 2014). The self-affine characterisation of subglacial topography described here informs
such techniques, and, in turn, could be used to inform the ice-sheet-wide interpolation of future
Greenland (Bamber et al., 2013a) and Antarctic (Fretwell et al., 2013) subglacial digital elevation
maps.

## 7 Summary and conclusions

In this study we used recent OIB RES data to demonstrate that subglacial roughness in northern
Greenland exhibits self-affine scaling behaviour, with pronounced spatial variation in the Hurst





(roughness power-law) exponent. We used a self-affine radar scattering model to predict how the Hurst exponent exerts control upon the spatial distribution of radar scattering, which we quantified using the waveform abruptness parameter. We then demonstrated a general agreement between the

predictions of the radar scattering model and the observed relationship between the Hurst exponent and waveform abruptness. Notably, high values of abruptness (specular reflections) are suppressed in regions of higher Hurst exponent, with continuous higher abruptness generally occurring in regions with a lower Hurst exponent. This agreement between predictions and observations enables us to conclude that self-affine roughness statistics provide a valuable framework in understanding the

topographic control which influences radar scattering.

     A central glaciological motivation behind our study was to establish if the waveform abruptness could be used to aid in the discrimination of thawed beds (under the initial assumption that thawed beds produce specular reflections). To do this we compared our RES-derived data fields with a predicted thermal state mask for northern Greenland. The analysis demonstrated that thawed regions

of the glacier bed have statistically lower values of the waveform abruptness than frozen regions. Hence, whilst high reflection specularity is a typical RES signature of subglacial lakes, it is not a typical RES signature of thawed regions of the Greenlandic subglacial interface. We therefore conclude that RES methods which seek to discriminate thawed regions, as distinct from deep (> 10 m) subglacial lakes or other electrically deep water bodies, should not use high waveform abruptness

(high reflection specularity) as a necessary criteria for discrimination of a thawed bed.

*Acknowledgements.* T.M.J., J.L.B., and C.N.W. were supported by UK NERC grant NE/M000869/1 as part of the Basal Properties of Greenland project. M.A.C. was supported by the UK NERC grant NE/L002434/1 as part of the NERC Great Western Four + (GW4+) Doctoral Training Partnership. We would like to thank J.A. Macgregor, NASA Goddard Space Flight Center, USA, for kindly supplying the Greenland thermal state mask.



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



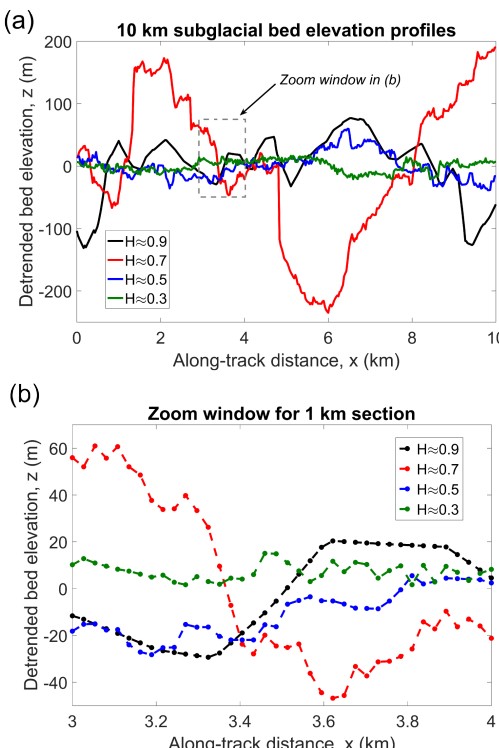

**Figure 1.** Examples of along-track bed elevation profiles for subglacial terrain with different Hurst exponent, $H$. (a) 10 km profile window. (b) Zoom to 1 km window with sample points indicated (corresponding to $\sim$ 30 m along-track resolution). The bed elevation is linearly detrended for each 10 km window. The aspect ratio differs between subplots.





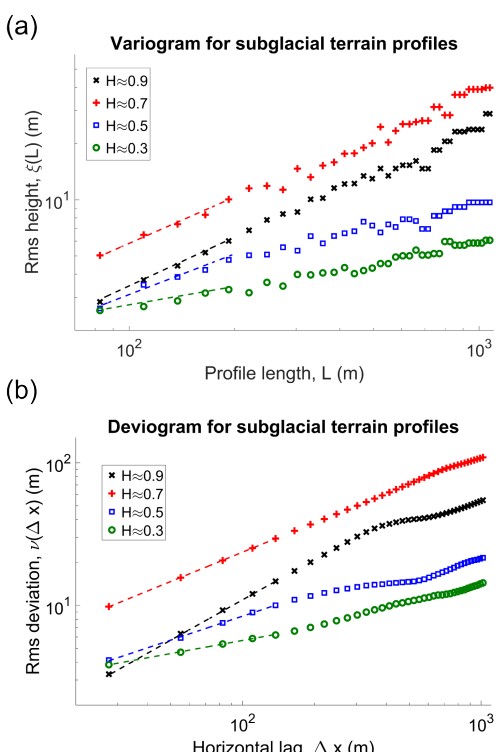

**Figure 2.** (a) Variogram for rms height, $\xi$, versus profile length $L$ (log-log scale). (b) Deviogram for rms deviation, $\nu$, versus horizontal lag, $\Delta x$ (log-log scale). The plots correspond to subglacial terrain profiles in Fig. 1a. The Hurst exponent is estimated from the linear gradient of the first five data points (indicated by dashed lines). These space-domain plots are (approximate) equivalents to frequency-domain roughness power spectra, and smaller length-scales correspond to higher frequencies.





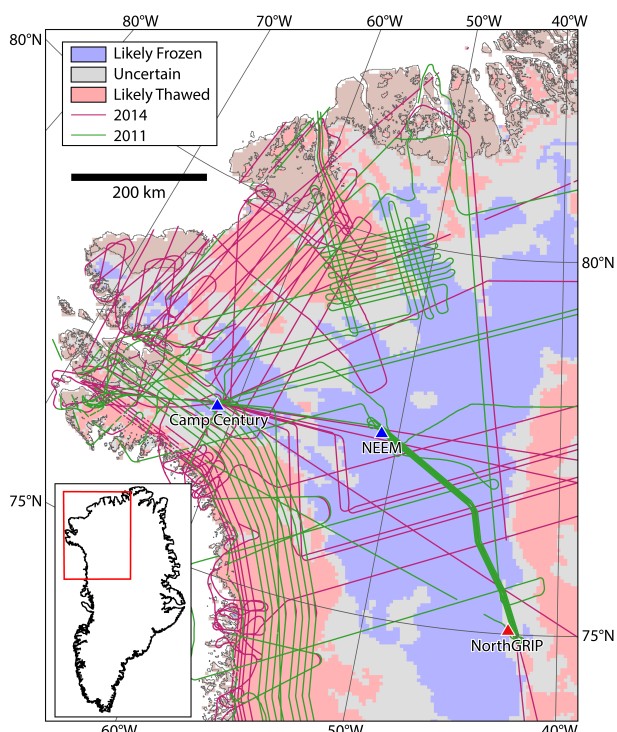

**Figure 3.** Data coverage map for OIB flight lines and region of interest. The map is underlain by the thermal state mask in Macgregor et al. (2016) where red, blue and grey colours correspond to predicted thawed, frozen and uncertain regions of the glacier bed. The locations of the Camp Century, NEEM and NorthGRIP ice cores are indicated.

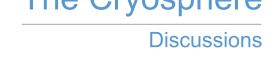

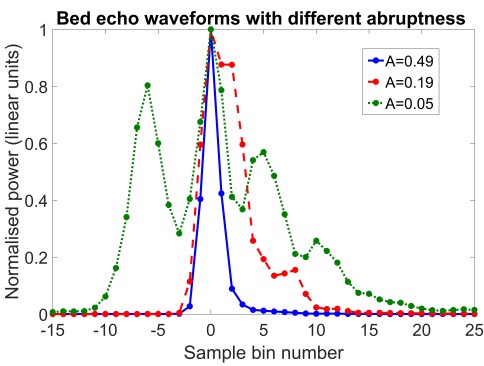

**Figure 4.** Examples of bed echo waveforms and their abruptness (pulse peakiness). Observed values for $A$ range from $\sim 0.03$ (associated with diffuse scattering) to $\sim 0.60$ (associated with specular reflection). For the purpose of comparative plotting, the waveforms are normalised about their peak power values with the sample bin of the peak power set to zero. The sample bin spacing corresponds to a depth-range spacing of $\sim 2.81$ m.



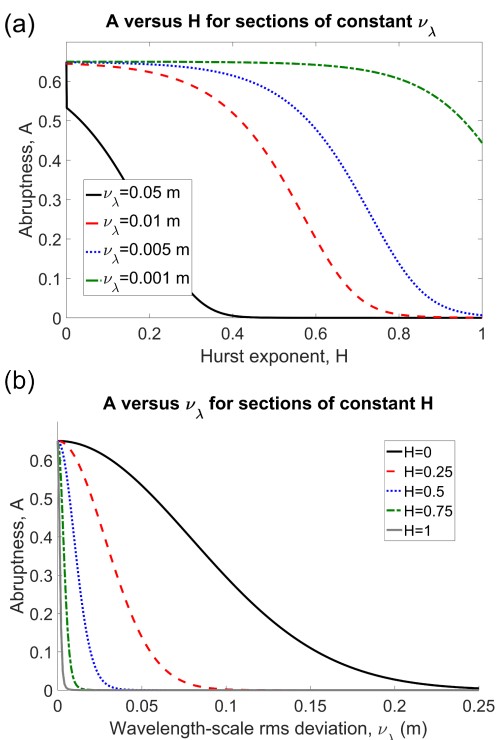

**Figure 5.** Parametric dependence of the self-affine radar scattering model. (a) Abruptness, $A$, as a function of the Hurst exponent, $H$, for sections of constant wavelength scale rms deviation, $\nu_\lambda$. (b) $A$ as a function of $\nu_\lambda$ for sections of constant of $H$. The plots illustrate primary dependence for $A$ upon $H$, and secondary dependence for $A$ upon $\nu_\lambda$. High $A$ is suppressed for high $H$, except in the case of exceptionally small $\nu_\lambda$.





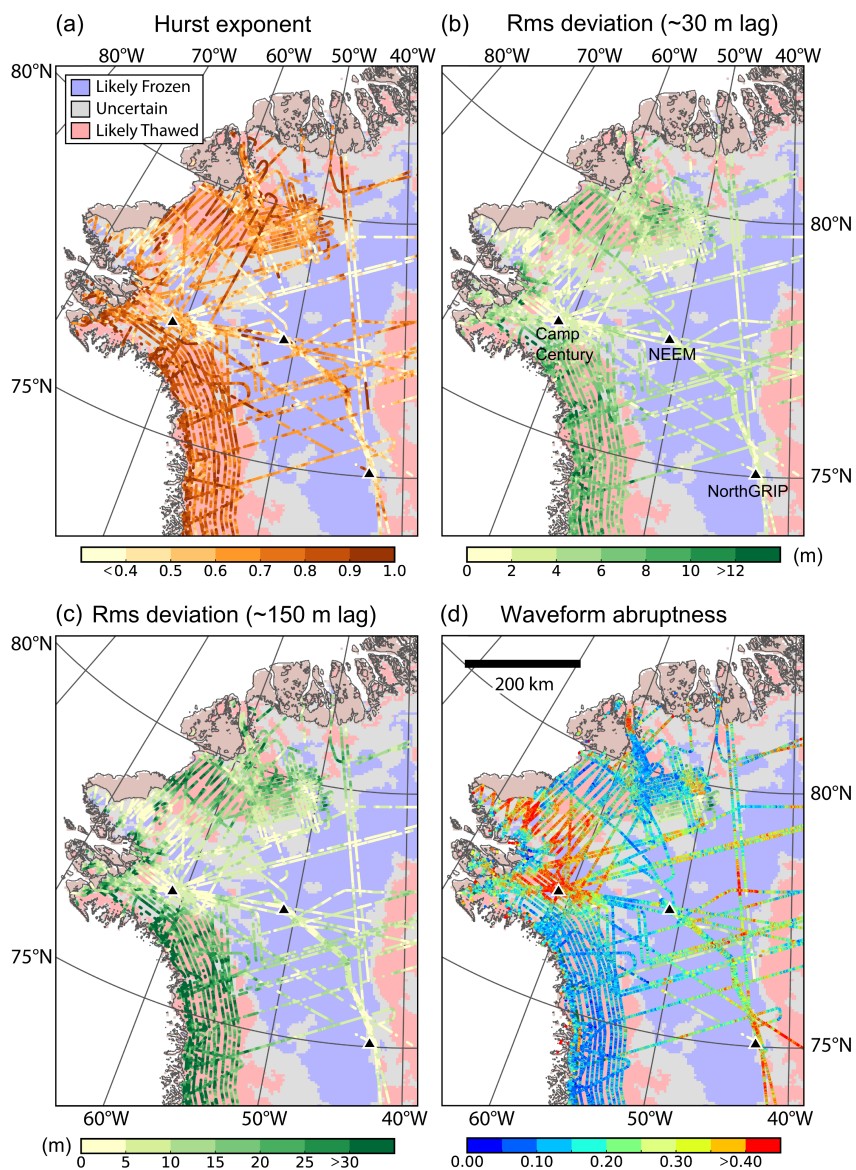

**Figure 6.** Data maps for basal RES analysis in the northern GrIS. (a) Hurst exponent, $H$. (b) Rms deviation (topographic roughness) at $\sim 30$ m lag, $\nu(\Delta x{=}30$ m). (c) Rms deviation at $\sim 150$ m lag, $\nu(\Delta x{=}150$ m). (d) Waveform abruptness (radar scattering), $A$. The maps are underlain by the predicted thermal state mask in Macgregor et al. (2016) where red, blue and grey colours correspond to predicted thawed, frozen and uncertain regions of the glacier bed. Higher values of $A$ in (d) indicate more specular reflections.





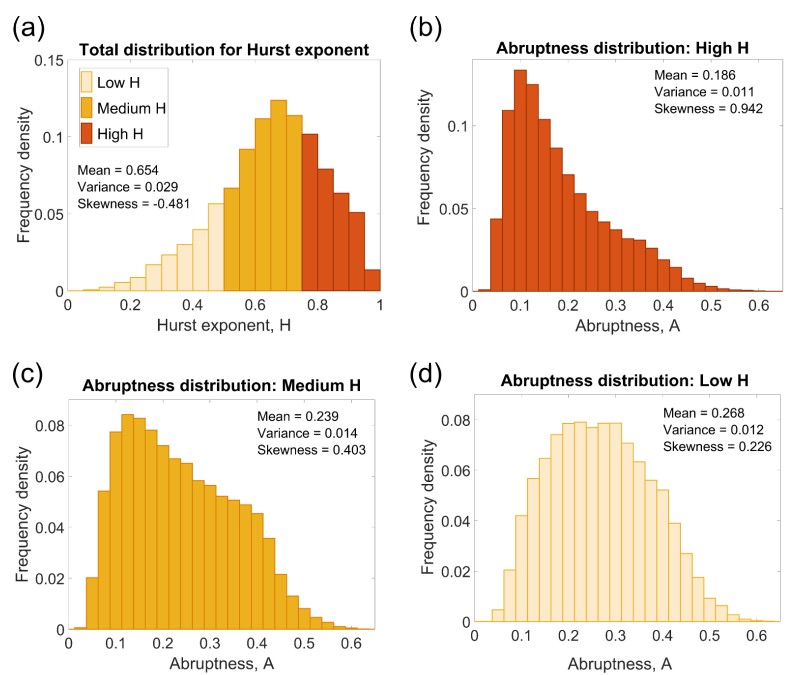

**Figure 7.** Relationship between Hurst exponent, $H$, and waveform abruptness, $A$. (a) Total distribution for Hurst exponent (corresponding to data in Fig. 6a). (b) Abruptness distribution for $H > 0.75$ ('High $H$'). (c) Abruptness distribution for $0.5 < H \leq 0.75$, ('Medium $H$'). (d) Abruptness distribution for $H \leq 0.5$ ('Low $H$'). The observed distributions in (b), (c) and (d) confirm the theoretical prediction of the self-affine radar scattering model that a statistically-distributed inverse relationship exists between $H$ and $A$. The analysis is performed independent of the thawed-frozen statistics in Sect. 5.3.





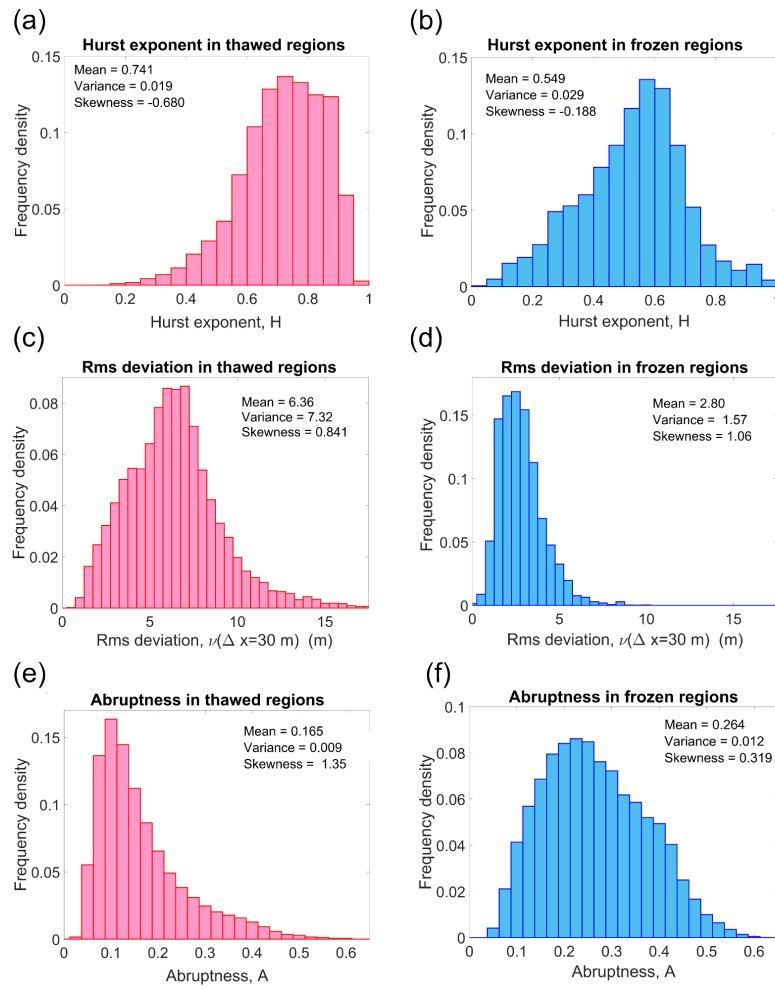

**Figure 8.** Distributions from basal RES analysis in thawed and frozen regions of the northern GrIS. (a) Hurst exponent, $H$, in thawed regions, (b) $H$ in frozen regions. (c) Rms deviation, $\nu(\Delta x=30$ m$)$, in thawed regions. (d) $\nu(\Delta x=30$ m$)$ in frozen regions. (e) Abruptness, $A$, in thawed regions. (f) $A$ in frozen regions. The data subsets correspond to the red (thawed) and blue (frozen) regions of the maps in Fig. 6.