# Peer review of "Self-affine subglacial roughness: consequences for radar scattering and basal water discrimination in northern Greenland"

_The Cryosphere, 2016_

## Referee Comment (RC1) · Anonymous Referee #1 · 16 Feb 2017

The ms demonstrates a theoretical relationship between the signal abruptness (a relative measurement of the signal decay) of the radar bed echo and the self-affine properties of roughness, the latter being a proxy for the thermal state of the bed. Then, a comparison of the signal abruptness with a thermal state map of the GIS bedrock illustrates how the signal abruptness can be used to outline at first order thawed and frozen bed.

The ms provides a valuable effort to introduce self-affine descriptors to classify glaciers' bed properties from radar return. The ms is well-written, the theoretical part is clearly shaped, and the discussion provides interesting insights into the qualitative relationship between bed properties and roughness as seen by radar. I have mainly few ques-

tions/minor revisions and technical remarks that, I hope, would enhance the quality of the ms.

General Questions

1) What kind of signal processing should be avoided to optimize your technique? For example, the waveform abruptness is mainly made from incoherent signal scattered by the surface. Does this mean that any coherent processing (coherent pre-summing or Doppler focusing) will tend to reduce the abruptness and its spatial contrast?

2) There is no mention of volume scattering in your ms while it is known to also contribute to the radar signal decay. What length scale for the heterogeneities of the volume scatterers would be needed to contribute to the abruptness? Could it explain some local mismatches between A and H on your maps? Is volume scattering a fair assumption in the context of bed and thawed glaciers' bed?

3) How do you choose the location of the bed echo in the case of the green waveform (A=0.05) on Fig.4? Do you compute $P_{agg}$? From fore and aft the chosen pick or just after it? Could you justify your choices and discuss putative bias arising from the specific case of this kind of waveform?

Specific Remarks

l.131: "proportional the rms deviation" should be "proportional to the rms deviation".

l.162-3: For clarity, the origin of the profiles should be moved in the first part of the paragraph (when you introduce the figure) and added in Fig.1 caption.

l.203-4: Could you briefly summarize the main signal processing applied to L1B data if any (any focusing or coherent pre-summing)?

l.240: "is also is consitstent" should be "is also consistent".

l.270: How long is the range window across wich you aggregate the power to get $P_{agg}$?

l.302: "can then obtained" should be "can then be obtained".

l.507-8: Specify that this H-A apparent correlation stands in the context of frozen/thawed glaciers bed. Your study does not show this relationship stands in other environments, espeially where volume scattering could be involved.

---

## Referee Comment (RC2) · N. Ross (Referee) · 16 Mar 2017

**Review of "Self-affine subglacial roughness: consequences for radar scattering and basal thaw discrimination in northern Greenland" (Jordan et al.,)**

The paper applies statistical techniques widely used in planetary radar sounding measurements to radar soundings of a terrestrial ice sheet for the first time. The authors demonstrate that a series of parameters (roughness, abruptness etc.) are different depending upon whether the data are from thawed or frozen bed conditions.

**General comments**

The paper is very well-written and technically excellent. The analysis is clearly important and has been very carefully and throughly undertaken. Some readers could perceive the manuscript as having a slight shortfall in glaciological content. However, I am of the opinion that the analysis and findings of the manuscript are of sufficient significance to make the content appropriate for publication in TC. For example, importantly, and of direct relevance to glaciology, the manuscript demonstrates limitations associated with the approach of Oswald and Gogineni (2008,2012) for determining areas of thawed and frozen bed. However, I do recommend that the authors do consider approaches to make the manuscript more accessible to a general glaciological audience. A slightly more glaciological-facing paper should acquire a broader readership in the community. I have two suggestions for how to achieve this, neither of which should be too onerous:

1. Incorporate into the paper a figure with radargrams that provide examples of the different characteristics. Including the actual radargrams for the four examples of the Hurst exponent in figure 1 could achieve this. Figure 4 might also benefit from radargrams illustrating bed echo waveforms with different abruptness.
2. What about a focused case study of a particular glacier (or two) in NW Greenland to exemplify the authors' key points? There are few references throughout the paper to actual locations or sites in Greenland, even in the discussion (e.g. Camp Century, NorthGRIP), and a few more references to geographical locations may make this paper a little less 'abstract' and more broadly accessible.

**Specific comments**
Intro – a very well-written intro, that sets the scene very effectively and concisely.

Section 3.2 – authors assume that all the level 2 data are consistent, but will have been picked by several different people. What confidence do the authors ascribe to the level 2 picks? Are there uncertainties associated with this data product that can be quantified?

Line 25 – are Seroussi and Schroeder references really the most appropriate here? I am sure that wet bed tends to = fast flow was determined by others long before 2013.

Lines 130- 131 – Sentence needs rewording.

Line 173 – some might suggest that reviewers could legitimately be referred to as 'asses', but the word I think you are looking for here is 'assess'.

Line 265 -"waveform corresponds to"?

Line 305 – "model a decrease"?

Line 400 – smoothing? A google search suggests "smoothening" is predominantly something to do with the management of hair.

Line 528 – "(Gorman and Siegert 1999)"

Line 529 – "…depth at which a lake is electrically 'shallow' or 'deep'….."?

Line 836 – "roughness"

Figure 6 – The colour scale in some parts of this figure (b and c) does not necessarily visualise the data as effectively as it might.

**Dr Neil Ross**
**Newcastle University**
**16th March 2017**
* * *

---

## Author Comment (AC1) · 13 Apr 2017

**Author comments from Tom Jordan, University of Bristol.**

We would like to thank both reviewers for their very thoughtful and constructive feedback, and welcome the opportunity to improve our manuscript. Below are our responses to both sets of reviewer comments, with our comments in blue text.

Anonymous Referee #1

The ms demonstrates a theoretical relationship between the signal abruptness (a relative measurement of the signal decay) of the radar bed echo and the self-affine properties of roughness, the latter being a proxy for the thermal state of the bed. Then, a comparison of the signal abruptness with a thermal state map of the GIS bedrock illustrates how the signal abruptness can be used to outline at first order thawed and frozen bed.

The ms provides a valuable effort to introduce self-affine descriptors to classify glaciers' bed properties from radar return. The ms is well-written, the theoretical part is clearly shaped, and the discussion provides interesting insights into the qualitative relationship between bed properties and roughness as seen by radar. I have mainly few questions/minor revisions and technical remarks that, I hope, would enhance the quality of the ms.

We now realise that a key message of our paper needs to be made explicit.  We are not proposing that we have a new RES diagnostic for basal water/thawed beds (either in terms of the Hurst exponent or the waveform abruptness). Rather; we provide evidence that the previously used RES diagnostic in Oswald and Gogineni (2008, 2012) (specular reflections for basal water) does not, in general, hold. Specifically; their approach will fail to identify water in rougher regions of Greenland (high H, Low A). Additionally, many of the contiguous regions of higher A that they argued to be thawed (such as around the Camp Century Core); are likely to be frozen.

Ultimately, we demonstrate that abrupt waveforms (specular reflections) are best viewed as being under geometric/topographic control (quantitatively expressed through the Hurst exponent), rather than providing a specific RES diagnostic for water or the thermal state.

We have therefore revised the way that we present the frozen/thawed analysis in our paper. Key changes are:

  (i)The abstract is now explicit about how our results impact upon basal water discrimination from RES, and the prior work of Oswald and Gogineni (2008, 2012)

  (ii)  The purpose of the frozen-thawed statistics (5.3) is now made clear throughout (i.e. testing the validity of Oswald and Gogineni (2008, 2012)).

  (iii) Fig. 6 and Fig. 3 now have the frozen-thawed underlay removed (the underlay is now a subplot in Fig. 6). This enables us to better focus on the topographic control theme.

On a related note, we are now clearer about our use of the term thawed bed (meaning regions above pressure melting point, as in Macgregor et al. 2016.), and the fact that the RES sounding method in Oswald and Gogineni (2008, 2012) was supposed to discriminate wet beds/basal water.

We have therefore revised the title to:

Self-affine subglacial roughness: consequences for radar scattering and basal **water** discrimination in northern Greenland

1) What kind of signal processing should be avoided to optimize your technique? For example, the waveform abruptness is mainly made from incoherent signal scattered by the surface. Does this mean that any coherent processing (coherent pre-summing or Doppler focusing) will tend to reduce the abruptness and its spatial contrast?

As a nadir-facing sounder the scattering contribution to the waveform abruptness is mainly from coherent reflection (as opposed to side-looking SAR instruments which would be mainly diffuse scattering). Whether coherent processing acts to increases or decrease the abruptness value depends on the exact character and roughness of the surface. For example, consider the two cases below:

Case A: The specular/nadir component is assumed to be coherent and delayed (abruptness decreasing) echoes, whilst the diffuse/off-nadir component is assumed to be incoherent (e.g. Grima et al. 2014). In this scenario coherent processing (either pre-summing of Doppler focusing) would cause the specular component of the signal to increase with coherent gain but not the diffuse (incoherent) signal. Therefore the measured abruptness would decrease with gain.

Case B: The specular (nadir) and diffuse (off-nadir) components of the echo are coherent (and therefore both experience coherent gain) (e.g. Schroeder et al. 2013, Schroeder et al. 2015). In this scenario, for small SAR processing angles (coherent pre-summing) the waveform abruptness should be unaffected and for larger angles (exceeding the angle spanned by the specular component of the echo in the scattering function) then the measured abruptness will decrease with inclusion of coherent processing.

We have now added a summary of this to Sect. 3.3, where the waveform abruptness is introduced.

2) There is no mention of volume scattering in your ms while it is known to also contribute to the radar signal decay. What length scale for the heterogeneities of the volume scatterers would be needed to contribute to the abruptness? Could it explain some local mismatches between A and H on your maps? Is volume scattering a fair assumption in the context of bed and thawed glaciers' bed?

Both our study (and to the best of our knowledge all other similar RES works) ignore Mie/volume scattering. We suggest this to an appropriate assumption for two reasons:

(i)      Ice-penetrating radar wavelengths, (~1-5 m in ice) are extremely long relative to ice surface remote sensing platforms (which we sense the reviewer may be more familiar with). Hypothetically, Mie/volume scattering would require a significant number of

obstacles of the order of this metre length-scale (or slightly smaller given the higher dielectric values in the bed). However, this is physically implausible given the micro to millimetre scale of typical water pore radii (e.g. Nimno2014), or the millimetre scale of heterogeneities in ice regoliths (e.g. Aglyamov2017).

(ii)      The radar signal in the bed often attenuates tremendously fast and echoes from any subsurface volume scatters (if hypothetically they did exist) would therefore not contribute significantly to the echo.

We have focused upon point (i) in the MS and have added the relevant references (see Sect. 4.1 when the radar scattering model is introduced).

3) How do you choose the location of the bed echo in the case of the green waveform (A=0.05) on Fig.4?

The position of the peak power was established by firstly using Level 2 Cresis picks, then applying a local re-tracker to centre over the peak power as show in Fig. 4. This has now been added to the text.

How do you compute Pagg? From fore and aft the chosen pick or just after it?

The integral considers fore and aft (since this better relates to the energy conservation arguments made later on), so we have now added this to the text. The calculation of P_agg was also described in more detail in our prior work in TC (Jordan et al. 2016), which we reference at the start of Sect 3.3.

Could you justify your choices and discuss putative bias arising from the specific case of this kind of waveform?

We think this is already alluded to in the text (in that our quality control filtering acts to filter out highly diffuse/messy waveforms). This would have the overall effect of slightly reducing our coverage in rougher regions, but is likely to be washed out by many of the other approximations we make in our paper: for example using a radially isotropic scattering model. We have therefore left the MS as is.

Specific Remarks

l.131: "proportional the rms deviation" should be "proportional to the rms deviation".

Changed as suggested.

l.162-3: For clarity, the origin of the profiles should be moved in the first part of the paragraph (when you introduce the figure) and added in Fig.1 caption.

Fig. 1 has now been substantially revised following the second reviewer's comments (including referencing the profile locations to our coverage map, Fig. 3).

l.203-4: Could you briefly summarize the main signal processing applied to L1B data if any (any focusing or coherent pre-summing)?

This has now been explained in 3.1.

l.240: "is also is consitstent" should be "is also consistent".

Changed as suggested

l.270: How long is the range window across which you aggregate the power to get Pagg?

The range window is actually variable (and relates to our power decay threshold described in 3.1). This should hopefully already be clear.

l.302: "can then obtained" should be "can then be obtained".

Changed as suggested

l.507-8: Specify that this H-A apparent correlation stands in the context of frozen/thawed glaciers bed. Your study does not show this relationship stands in other environments, especially where volume scattering could be involved.

See our previous comments on volume scattering.

Discussion

Author Comments 2

Review of "Self-affine subglacial roughness: consequences for radar scattering and basal thaw discrimination in northern Greenland" (Jordan et al.,)

The paper applies statistical techniques widely used in planetary radar sounding measurements to radar soundings of a terrestrial ice sheet for the first time. The authors demonstrate that a series of parameters (roughness, abruptness etc.) are different depending upon whether the data are from thawed or frozen bed conditions.

General comments

The paper is very well-written and technically excellent. The analysis is clearly important and has been very carefully and thoroughly undertaken. Some readers could perceive the manuscript as having a slight shortfall in glaciological content. However, I am of the opinion that the analysis and findings of the manuscript are of sufficient significance to make the content appropriate for publication in TC. For example, importantly, and of direct relevance to glaciology, the manuscript demonstrates limitations associated with the approach of Oswald and Gogineni (2008,2012) for determining areas of thawed and frozen bed. However, I do recommend that the authors do consider approaches to make the manuscript more accessible to a general glaciological audience. A slightly more glaciological-facing paper should acquire a broader readership in the community. I have two suggestions for how to achieve this, neither of which should be too onerous:

1. Incorporate into the paper a figure with radargrams that provide examples of the different characteristics. Including the actual radargrams for the four examples of the Hurst exponent in figure could 1 achieve this. Figure 4 might also benefit from radargrams illustrating bed echo waveforms with different abruptness.

This is an excellent suggestion and Fig 1 (along-track bed elevation profiles) has been revised to include example radargrams. Underneath each radargram, each sub-figure bed-elevation profile now has a different aspect ratio which allows the reader to better visual surfaces with different H (in particular the persistence/anti-persistence of the elevation trends). The location of the profiles are now added to the coverage map (Fig. 3); which will enable readers to place profiles in the context of the Northern GrIS.

2. What about a focused case study of a particular glacier (or two) in NW Greenland to exemplify the authors' key points? There are few references throughout the paper to actual locations or sites in Greenland, even in the discussion (e.g. Camp Century, NorthGRIP), and a few more references to geographical locations may make this paper a little less 'abstract' and more broadly accessible.

This is again very helpful feedback and we have given the results and discussion an edit to make it more accessible to a geographical/glaciological audience.

Key changes include:

(i) Section 5.1 (the flight-track maps) now draws a comparison with Greenland bed topography. This comparison enables us to better place the RES-derived data (Hurst exponent, Waveform Abruptness, RMS deviation) in its geographical context. Specific locations have also been added to Fig, 6 and are referred back to in the discussion.

(ii) The uncertainty/sensitivity analysis (previously included in 5.1) has now been moved to its's own sub-section (5.4); thus improving the overall flow of the geographical results.

(iii) The discussion has been re-ordered, with more specific geographical locations and context included.

Specific comments

Intro – a very well-written intro, that sets the scene very effectively and concisely.

Section 3.2 – authors assume that all the level 2 data are consistent, but will have been picked by several different people. What confidence do the authors ascribe to the level 2 picks? Are there uncertainties associated with this data product that can be quantified?

The picking procedure is now described (in 3.1 rather than 3.2), outlining that we use the highest confidence picks from the Cresis data produce.

It is actually quite complex what uncertainty metric is relevant for our statistical analysis of along-track bed-elevation. For example, the presentation of Level 2 picks in Gogineni et al., 2001 and Bamber et al. 2013 subscribes an uncertainty ~ 10 m (based on cross-over analysis of flight-tracks). This, however, considers separate tracks, and is likely an overestimate for our along-track statistical

application. Additionally, the manual picking preclude would introduce error auto-correlation (i.e. neighbouring points would likely have a similar bias). We therefore believe that the theoretical depth-range value (which we already quote) is the most useful number (even if it does not account for the picking procedure.)

Line 25 – are Seroussi and Schroeder references really the most appropriate here? I am sure that wet bed tends to = fast flow was determined by others long before 2013.

We have now added Weertman's and Nye's classic works on basal sliding.

Lines 130- 131 – Sentence needs rewording.

Changed to: `… and we focus upon this roughness parameter when integrating topographic-scale roughness with radar waveform data.'

Line 173 – some might suggest that reviewers could legitimately be referred to as 'asses', but the word I think you are looking for here is 'assess'.

Done.

Line 265 -"waveform corresponds to"? #

Done.

Line 305 – "model a decrease"?

Done.

Line 400 – smoothing? A google search suggests "smoothening" is predominantly something to do with the management of hair.

Done.

Line 528 – "(Gorman and Siegert 1999)"
Done.

Line 529 – "…depth at which a lake is electrically 'shallow' or 'deep'….."?

Sentence has been re-written.

Line 836 – "roughness"

Done.

Figure 6 – The colour scale in some parts of this figure (b and c) does not necessarily visualise the data as effectively as it might.

The colour scale has now been revised. We also have removed the frozen/thawed underlay (instead having it as a subplot), which improves the colour contrast between the flight-track data and the background.

Dr Neil Ross
Newcastle University
16th March 2017
* * *

---

## Author Response (AR1)

Manuscript prepared for The Cryosphere
with version 2015/04/24 7.83 Copernicus papers of the LATEX class copernicus.cls.
Date: 13 April 2017

**Self-affine subglacial roughness: consequences for radar scattering and basal thawwater discrimination in northern Greenland**

Thomas M. Jordan[1], Michael A. Cooper[1], Dustin M. Schroeder[2], Christopher N. Williams[1], John D. Paden[3], Martin J. Siegert[4], and Jonathan L. Bamber[1]

[1]Bristol Glaciology Centre, School of Geographical Sciences, University of Bristol, Bristol, UK.
[2]Department of Geophysics, Stanford University, Stanford, California, USA.
[3]Center for Remote Sensing of Ice-Sheets, University of Kansas, Lawrence, Kansas, USA.
[4]Grantham Institute and Department of Earth Science and Engineering, Imperial College, London, UK.

*Correspondence to:* T. M. Jordan (tom.jordan@bris.ac.uk).

*The abstract been revised. In particular, we are explicit about how our study impacts upon the prior work of Oswald and Gogineni (2008, 2012). The rest of the abstract has also been edited to make room for the extra content and for improved accessibility.*

[revised manuscript text omitted]

*We have removed the frozen-thawed mask from the data coverage map (and moved the related text to the results. Extra information on L1B pre-processing has also been added here.*

[revised manuscript text omitted]

280 point was identified.

**3.3 Determination of waveform abruptness from Level 1B data**

*We have now added extra information about how L1B pre-processing relates to the abruptness signal.*

[revised manuscript text omitted]

*Following the reviewer comments the presentation of the results section has been substantially revised (although we would like to stress that none of the results are actually any different than before). Key changes include: (i) Comparing the RES-derived data with the DEM and broadening the geographical interpretation of the data maps in 5.1. (ii) placing the uncertainty analysis at the end of the results (thus making the presentation more accessible), (iii) Better separating the presentation of topographic and thermal control of radar scattering.*

[revised manuscript text omitted]

*The next section has been removed since it adds little to the overall results or conclusions.* ~~We repeated the above analysis for $H$ values that were estimated using the variogram for $\xi(L)$. The distributions for the three $H$ categories are qualitatively similar to Fig. 7b-d but with less pronounced quantitative differences between $H$ categories (e.g. the mean values for $A$ are 0.201, 0.243 and 0.254 for High, Medium and Low $H$ categories respectively). This is as expected, since the length-scales at which $H$ is estimated using the deviogram are more directly related to radar scattering than the variogram.~~

**5.3 Statistics in thawed and frozen regions**

*To better introduce the frozen-thawed statistics we have now rewritten the first paragraph (by moving some of the material about the O+G algorithm that was originally in the discussion to this section)*

[revised manuscript text omitted]

*This has been moved to the radar scattering model section* ~~It is important to note that both the predictions of the self-affine radar scattering model (Sect. 4), and the observed relationship between the abruptness and Hurst exponent (Sect. 5.1, Sect. 5.2), are consistent with the specular RES scattering signature that we would expect from electrically deep subglacial lakes. Under the self-affine roughness framework, a large geometrically flat feature such as a lake would have a negligible value of $H$ and $\nu_\chi$. This scenario occurs for the low $H$ limit of the green curve in Fig. 5a, where predicted values for $A$ are $\sim 0.65$ (corresponding to a perfectly specular reflection).~~

*The following paragraph has been integrated with others in the discussion (some of the conent also repeated the parts of the introduction and was therefore redundant*

**7  Summary and conclusions**

*We have performed a general edit and re-emphasised the significance of our study for basal thaw discrimination*

[revised manuscript text omitted]

---

## Author Response (AR2)

We would like to thank our editor, Olaf Eisen, for his thorough and helpful comments which have further improved our manuscript. Our responses are below in blue text.

Editor comments

line 136: delete one "where"

Done

249-251: rewrite last sentence, difficult to understand

Changed to:

`Finally, break points in the linear relationship were identified by testing if the new gradient exceeded a specified tolerance from the original estimate.'

281: "glacial" does not fit, I suggest to change to "As RES over ice employs …"

Done

315: scatterer dimensions are …: unclear what the numbers mean, relative dimension of dimension of space (then unit is missing)

Good spot. Units of m have been added

334: split sentence: assumes radial isotropy for H and Xi. Since we are …

Done

338: repetition: Paden already cited above where wavelength is first introduced.

Done

378: Unclear where 0.65 is coming from and what it means here: resolution of number of range cells? If resolution, why? Unit missing?

A good point. We have now been clearer about how we arrive at C~ 0.65:

`For a perfectly specular reflection the pulse is the shape of compressed chirp (absolute value of a sinc function with the width determined by the signal bandwidth). If the depth-range sampling of the waveform (Fig. 4) were the same as the depth- range-resolution then C would be near unity. However, since the depth-range sample spacing (~ 2.8 m) is less than the depth-range resolution (~ 4.3 m), C can be estimated from the number of depth-range cells which fit in the depth-range resolution C ~ 0.65.'

413: (5.2) -> (Sect. 5.2)

Done
460: although it clear that: add "is"#

Done

516: 100s km2 -> 100s of km2

Done

685: remove () from citation

Done

Figure 1: last sentence: fix grammar "have … is". Add after last sentence: by a factor of 10.

Done

Figure 6: title and first sentence in caption: northern Greenland -> north-eastern Greenland

Much of our data map and discussion also includes North-central Greenland (e.g. NGRIP). We therefore feel that 'northern Greenland' is more a more encompassing descriptor.

Figure 7: I suggest to use the same scale for y-axes of b, c, d at least, as not only the shape but also the area is important for visual comparison.

Figure 8: Same as for Fig. 7: E.g. c and f look dominant, which is only attributed to the different y-scale

We respectfully wish to keep the y-axis scales as they are. The area of each histogram is self-normalised and it is the relative shape (quantified by mean values, skewness, variance) and the x-axis scale (which is fixed between plots) that is important for the arguments we make in the paper. Additionally, if the y-axis scales are identical then there is an unnecessary large amount of white space for some subplots (and in general the plots are less aesthetically pleasing).

Typesetting:
- unify usage of % without leading space, i.e. 1% not 1 %

Done

- l 309: delete "," after scattering

Done

- 418: "data, " -> "data "

Done

- 576: "surface, " -> "surface "

Done

- 615: thickness – correlated -> thickness-correlated

Done

- Fig. 6: panel titles and caption: Rms looks unusual. The text uses RES and rms. I suggest to unify by writing RMS throughout the text or at least using RMS here.

Last time we published in TC, we were advised to use rms in the text presentation of root mean square (which is what we have done here). We have therefore amended the figures to also use rms rather than Rms (apologies – this was done for consistency with capitalisation of other fig. labels).

- several very minor aspects are not conform with the typesetting instructions and will be dealt with during copyediting, e.g. usage of multiple braces for references, hyphenations, etc.